# Towards resilience: Transcriptional insights on flavonoid biosynthesis during peanut seed maturation phases

Gustavo Roberto Fonseca de Oliveira[1]◉*, Liam Walker[2]‡, Rômulo Pedro Macêdo Lima[1]‡, Thiago Barbosa Batista[3]‡, Clíssia Barboza Mastrangelo[3]‡, Edvaldo Aparecido Amaral da Silva[1]◉

1 School of Agronomy Science, São Paulo State University, Botucatu, São Paulo, Brazil, 2 School of Life Sciences, University of Warwick, Coventry, United Kingdom, 3 Laboratory of Radiobiology and Environment, Center for Nuclear Energy in Agriculture, University of São Paulo, Piracicaba, São Paulo, Brazil

◉ These authors contributed equally to this work.
‡ These authors also contributed equally to this work.
* grfonseca.agro@gmail.com

## Abstract

Flavonoids are secondary metabolites widely studied as a metabolic protector against several stressors in plants, yet they are understudied in the events of peanut seed development. Substantial advances in the understanding of peanut seed maturation have been made in recent years, however, the role of flavonoids in this process is unclear. Here, the fundamental question asked was: are flavonoids involved in peanut seed maturation phases? This study investigates whether the main transcripts associated with the flavonoid pathway, such as anthocyanins, are biologically linked to the physiological quality components of peanut seeds during development. For this purpose, peanut seeds classified into five stages (R5, R6, R7, R8 and R9) were used for the evaluation of quality attributes, such as desiccation tolerance, vigor and longevity parameters, and for RNA sequencing (RNA-seq). Interestingly, anthocyanins accumulated more in the beginning of the seed filling phase, coinciding with the expression upregulation in RNA-seq and quantitative PCR of key genes in its pathway, such as *AhCHS* (0FI6RG), *AhCHI* (VJQ7J1), *AhFLS* (4Y1607), *AhLDOX* (AQ6B1J) and *AhANR* (IK60LM). Additionally, we found that *AhMYB12* (PR7AYB) and *AhMYB308* (TG6F30) exhibited increased expression at the early stages (R5 and R6) and decreased at the later ones (R7, R8 and R9). The *AhCHS* expression acts in synergy with well-known seed maturation regulators, such as the ABA response (e.g., *ABI5* and *ABI3*). The involvement of flavonoid biosynthesis in peanut seed development is suggested here as a contributor to its resilience during the acquisition of physiological quality attributes, highlighting the molecular aspects associated with survival in the dry state.

**Data availability statement:** All relevant data are within the manuscript and its Supporting Information files.

**Funding:** The authors G.R. Fonseca de Oliveira, E.A. Amaral da Silva and C.B. Mastrangelo are also grateful to the National Council for Scientific and Technological Development (CNPq; grant nº 408623/2024-1, grant nº 311526/2021-7, grant nº 142236/2020-9 and grant nº 314305/2021-1) and the São Paulo Research Foundation (FAPESP; grant nº 2018/03793-5, 2020/14050-3, 2021/07331-9 and, 2023/00435).

**Competing interests:** The authors have declared that no competing interests exist.

## Introduction

Flavonoids comprise an important group of secondary metabolites with an antioxidant role that has been broadly investigated in plants [1,2]. These compounds are generally present in the cell vacuole and they are widely found in structures such as leaves, flowers, fruits and seeds [3]. Depending on the plant species and organ, different types of flavonoids can help in combating pathogens, insect attacks and excess free radicals [4,5]. Anthocyanin, in addition to these functions, has been described as a pigmenting flavonoid that mitigates oxidative stress in plant tissues, ensuring cellular homeostasis under biotic and abiotic adversities [6,7]. In dry seeds, anthocyanin participates in an autoimmune system that reacts to aging and exposure to thermal stress, with protective effects that extend to the seedlings [8,9]. This represents an interesting embryonic resilience strategy that, like many others, is generally acquired during seed development and is associated with physiological changes and established molecular mechanisms [10,11]. Considering this, we investigated whether the transcription dynamics of flavonoids, such as anthocyanin, could be involved in with the acquisition of seed quality attributes during the maturation phases.

Seeds complete their development in an ordered maturation program subdivided into a time series marked by seed filling/maturation and what we recognize nowadays as late maturation [12,13]. The maturation phase of desiccation-tolerant seeds (orthodox) is primarily characterized by abrupt changes in water status and weight [14,15]. The seeds undergo a constant accumulation of reserves components such as proteins, sugars and oil, in addition to high hormonal/metabolic activity and molecular regulation [16–18]. It is an "orchestra" of agents that work to ensure the preservation of seed life during desiccation and storage [19,20]. In the late maturation phase, the accumulation of reserves is completed and the seeds continue to lose water and consolidate cellular stability towards the acquisition of vigor and longevity [12,13]. These components allow the seeds to do something extraordinary in nature: germinate and generate seedlings when hydrated after a prolonged dry state [11,21]. This is extremely important for agriculture and food security, since seeds harvested in one season need these physiological resources to generate high-performance plants in the following production cycle [22–24]. Therefore, the fundamental question to be answered is: what is the collaborative part of flavonoids, such as anthocyanin, in this "orchestra" programmed for the acquisition of components necessary to survive the dry state? Flavonoids are widely studied in the context of plant antioxidant defense but the potential contribution of flavonoids in the context of maturation phases has been very little discussed [5,25,26]. This is even less investigated in peanuts, an oilseed of great relevance for global agriculture [27,28].

The flavonoid biosynthesis pathway is widely known [1,4]. The main associated enzymes include chalcone synthase (CHS), chalcone isomerase (CHI), flavonol synthase (FLS), leucoanthocyanidin dioxygenase (LDOX) and anthocyanidin reductase (ANR) [5,29]. Additionally, proteins of the MYB family transcription factors often activate or repress the expression of the flavonoids genes [30,31]. For example, MYB12 directly induces the synthesis of different flavonoids according to the biological demand

for protection in plants [32,33]. On the other hand, MYB308 represses this synthesis but enriches other agents of the plant secondary metabolism pathways in return, modulating enhanced self-immunity [34–36]. Therefore, these transcription factors regulate the expression of genes from a "powerful toolkit" associated with protection [34–36]. Interestingly, other key players associated with seed maturation phases have a similar protective role in the molecular level [10,21]. Several heat shock proteins (HSPs), late embryogenesis abundant proteins (LEAs) and oligosaccharides from the raffinose family (RFOs) coordinate the protection of seeds during desiccation and the acquisition of longevity [10,16,37]. In particular, proteins with chaperone activity, such as HSP70, play a key role in guiding other proteins to fold correctly into their functional three-dimensional shapes, ensuring that they operate efficiently, especially during stress conditions [38,39]. These events are also regulated by transcription factors such as those associated with the ABA response (e.g., *ABI5* and *ABI3*), which have crucial functions in seed maturation [12,15,40,41]. Thus, we explored the possible links between flavonoids and these coordinators of peanut seed development. The hypothesis tested was that flavonoids are transcriptionally involved in peanut seed development, acting systematically on secondary metabolism throughout the maturation phases.

## Materials and methods

### Seed samples

Peanut seeds of the Runner type belonging to the cultivar IAC OL3 were used. The seeds were produced at the Experimental Farm Lageado of São Paulo State University, Botucatu, São Paulo, Brazil, in the 2021/2022 season. The material was chosen considering its high yield and significant adoption by Brazilian farmers. The field conditions for crop management, including soil preparation, fertilization (top dressing) and weather variations during the plant cycle (rainfall, temperature and relative humidity, RH), were described (2021/2022 crop season) [24]. Additionally, following the methods presented by those authors, the seeds were collected in the field, classified according to their development stages (R5, R6, R7, R8 and R9) and a portion of them was dried until it achieved 10% of water content. The samples were used to conduct physiological assessments. For the molecular analyses, such as RNA sequencing (RNA-seq) and quantitative PCR (qPCR), we use fresh seeds immediately frozen after harvesting. The details of which are presented below.

### Seed physiology and anthocyanin

**Water and weight.** The water determination took place by using five replicates of ten fresh seeds (n = 50) from each development stage. The material was put into a cylinder container with a lid and weighed on a precision balance (accurate to 0.0001 g). The seeds were kept at 105ºC for 24 hours (oven method) [42]. The water content was expressed in humidity degree on a wet basis (%). For the dry weight determination, another five replicates of ten seeds were used (n = 50). Similar containers were adopted and the seeds were kept at 60ºC for 72 hours. The dry weight was expressed in grams per 10 seeds [24].

### Desiccation tolerance

Five replicates of 25 dry seeds from each development stage were used (n = 125). The method for peanut seed coat removal and sample sterilization (1% hypochlorite) was followed as described [24]. The seeds were placed on paper rolls moistened with deionized water (1:2.5, g:mL). The material was put in an incubator set to 20ºC for 12 hours (lights off) and 30ºC for another 12 hours (lights on), as described [43]. The desiccation tolerance (%) was assessed 10 days afterwards, considering radicle protrusion ≥ 2 mm as germinated seed criterion.

### Time to 50% Germination (t50)

For the t50 test, five replicates of 25 dry seeds were used (n = 125). The procedures to promote seed germination described above were adopted as well [24]. Germination, expressed by radicle protrusion ≥ 2 mm, was evaluated

every 4 hours from the start of the test. The software Germinator was used to calculate the t50 according to the seed stages [44]. The t50 was expressed in hours, where more time to reach 50% germination indicates lower seed performance.

### Seedling assessments

For the analysis of seed vigor through seedling performance, five replicates of 10 dry seeds from each development stage were used (n = 50). The seeds were placed in paper rolls, in the upper third, with the hilum facing downwards. The paper was moistened as previously (1:2.5, g: mL) and they were placed in a controlled incubator (20ºC for 12 hours with no light/ 30ºC for 12 hours with light). After 7 days, the total length of the seedlings was measured (cm). Additionally, the shoots and roots were separated to determine the dry weight, expressed in milligrams (60ºC for 72 hours) [24].

### Germination of aged seeds

For each development stage, 145 dry seeds were subjected to the aging method [45]. The seeds were placed on a wire mesh, set inside a plastic box containing 40 mL of water and kept at 42ºC for 3 days. After aging, seed coat removal and sterilization procedures were adopted, as described [24]. Germination was performed using five replicates of 25 seeds (n = 125). Two replicates of 10 seeds were used for the water content determination, as previously described. The germination of aged seeds was obtained after 5 days, considering radicle protrusion ≥ 2 mm as criterion.

### Seedling emergence

Five replicates of 25 dry seeds from each development stage were used (n = 125). The seeds were sown in a sand compost equidistantly and under field conditions. The sand was watered to 60% of its retention capacity. The weather variables were monitored during the test as described [24]. The criterion for counting seedling emergence (%) was the appearance of shoots above the sand with expanded leaves by the 21st day. The speed index was calculated based on the sum of daily counts of seedlings over time (days), as described [46,47].

### Plant establishment in the field and shoot dry weight

Five replicates of 20 dry seeds (n = 100), stored for 6 months (10ºC and 55% RH), were used. The seeds were sown in the soil in five parallel lines for each development stage (1m, 90 cm apart), one line per replicate. No chemical treatments were applied on the seeds so that their natural self-defenses under field conditions could be evaluated. Water supplementation was necessary (15 mm, sprinkling) to ensure basic moisture for seed germination. Plant establishment (%) was counted on the 30th day after sowing. Finally, all the plants in the lines (plant stand) had their shoots cut off for dry weight assessment (60ºC for 72 hours, milligrams).

### Germination of stored seeds

To check their storage life, 145 dry seeds at each development stage were used. The seeds were arranged on a piece of gauze cloth, which was fastened to the top surface of a container, as described [24]. The container's bottom was filled with a saturated NaCl solution. After sealing the box tightly, it was placed in an incubator at 35ºC and 75% RH for 100 days. Germination was performed using five replicates of 25 seeds (n = 125). Additionally, water content was determined (around 7%) using two replicates of 10 seeds, as described previously. Radicle protrusion ≥ 2 mm was considered as the criterion for germination (%) 10 days after the beginning of the test. Seedlings with well-formed shoots, fully developed roots and total length ≥ 3 cm were considered normal (%). The results expressed the capacity of the seeds to tolerate storage, keeping their viability under stress conditions (high temperature and relative humidity).

## Anthocyanin and multispectral images

Five replicates of 15 dry seeds from each development stage were used (n = 75). Multispectral images were captured non-destructively with a SeedReporter™ (PhenoVation, Netherlands) to calculate the anthocyanin index, as described [9]. Light intensity was adjusted to prevent overload and reflectance images (2448 × 2448 pixels, 3.69 μm/pixel) were acquired under broad-band white light (3000 K, 450–780 nm) using filters at 540, 710 and 770 nm. The anthocyanin index was computed with SeedReporter™ software (v5.5.1) following described procedures [48].

## Seed molecular assessments

**RNA extraction.** Three replicates of 15 fresh seeds (n = 45) were frozen in liquid N immediately after they had been collected at each development stage. Seed samples were stored in a deep freezer at −80ºC. Each replicate was manually crushed in a crucible with a pestle until it reached a powdery appearance. This procedure was always performed in the presence of liquid N. The crushed material was placed into 1.5 mL tubes and stored at −80ºC, as described [49]. For RNA extraction, the NucleoSpin RNA Plant® Kit from Macherey-Nagel was used according to the manufacturer's instructions. RNA purity was determined by UV spectrophotometry (260 nm, NanoDrop 1000, Thermo Fischer Scientific) and its integrity was determined using a Bioanalyzer, Agilent Technologies (Life Sciences Core Facility, LaCTAD, State University of Campinas, UNICAMP) (https://www.lactad.unicamp.br/). Only samples with an RNA Integrity Number (RIN) > 7 were used for RNA sequencing (RNA-seq), as shown in the supplemental material (S1 Table in S1 File).

### RNA-seq and bioinformatics

For cDNA library construction, the RNA samples from 15 fresh seeds of each development stage were pooled. Three biological replicates per stage were used. The independent sequencing from RNA-Seq was performed with the following parameters: Illumina platform, HiScan; 150 bp paired-end and with a number of lanes equal to two; mean scores ≥ 20 per position were used. Raw reads were trimmed using fastP [50] with the following parameters; cut_right, cut_window_size = 4, cut_mean_quality = 20 and length_required = 40. Alignment of trimmed, clean reads and quantification of counts per gene was conducted using STAR [51] with the sjdbOverhang parameter set to 149 (equal to read length −1). The index files for STAR were generated using the *Arachis hypogaea* accession (arahy.Tifrunner.gnm1.ann1.CCJH.gene_models_main.gff3.gz) Tifrunner annotation files from genome assembly 1.0 [52]. Mapping reads of the peanut genome are available in supplemental information (S2 Table in S1 File) and the less recent annotation file from genome assembly 1.0 was chosen due to compatibility with the Phytozome database [53] (https://phytozome-next.jgi.doe.gov/). Differentially expressed genes were obtained using DESeq2 [54]. Changes in gene expression were considered statistically significant when $Log_2$ Fold Change (LFC) was ≤ −2 and ≥ 2. The differentially expressed genes (DEGs) were obtained with adjusted *p*-values cutoff ≤ 0.001. The key molecular players of seed maturation phases were highlighted in the DEGs data considering previous studies [10,11,15,16].

## Gene Ontology (GO) and ortholog identification

The GO enrichment analysis was carried out through the platform agriGO v2.0 [55]. Significantly differentially expressed genes were queried for overrepresented ontologies against the *Arachis hypogaea* cv Tifrunner (*Ah*) database (http://systemsbiology.cau.edu.cn/agriGOv2/). Amongst the most enriched processes was flavonoid biosynthesis which was identified with a false discovery rate (FDR) < 0.000001. To identify orthologs of characterized flavonoid biosynthesis genes from other species [6,32], the protein homologs tool of Phytozome 13 (https://phytozome-next.jgi.doe.gov/) and NCBI BLAST (https://blast.ncbi.nlm.nih.gov/Blast.cgi) were used. The normalized gene count data (DESeq2, rlog) was then used to obtain the expression of these genes from our data (three replicates of 15 fresh seeds, n = 45).

### RT-qPCR study

To validate the differentially expressed genes associated with anthocyanin synthesis identified in the RNA-seq, an RT-qPCR study was performed. For this, primers for the target genes were designed (S3 Table in S1 File) based on the nucleotide sequences of the corresponding transcripts, using the Primer 3 platform (https://primer3.ut.ee/) [56]. The same total RNA sequenced in three replicates for each development stage (R5, R6, R7, R8 and R9) was used. cDNA synthesis was performed using the High-Capacity cDNA Reverse Transcription Kit (Thermo Fisher). The amplification protocol consisted of 3 minutes at 95ºC, followed by 40 cycles of 15 seconds at 95ºC and 1 minute at 60ºC and a final step of 1 minute at 95ºC and 1 minute at 55ºC. The $2^{-\Delta\Delta Ct}$ method was used [57], where $\Delta\Delta Ct$ represents the normalized expression of the target gene against the reference gene, 60S ribosomal protein L10 described in peanut studies [58,59]. The expression of the genes during seed development was calculated relative to a pool of all seed stages.

### Data analysis

The experiment involved randomly assigning five development stages (R5, R6, R7, R8 and R9) as treatment groups, with five independent replicates for each stage (n = 125). Mean differences among the groups were determined through the Scott-Knott test, applying a significance threshold of $p$ value ≤ 0.05. Additionally, The PCA made with seed physiology and anthocyanin data was presented with a PERMANOVA test performed using Canoco 5 software [9,60]. Correlation analyses were carried out using the Spearman method. Finally, Mean Decrease Gini (MDG) was obtained with a random forest algorithm, where variables with higher MDG values are key genes, which are the predictors of seed development stages [24]. The data analysis was performed using R software [61].

## Results

### Physiological quality and anthocyanin in peanuts

The basic events of developing seeds followed patterns with sequential logic; water content decreased between stages R5 and R9 (70−30%), coinciding with an increase in dry weight (Fig 1A-B). The filling phase occurred up to R7, the stage at which the seed began to tolerate desiccation and express viability in the dry state (63.2%) (Fig 1C). Even with stable weight (7.58 g/10 seeds), the seeds continuously enhanced their capacity to germinate quickly and establish as seedlings after storage (Fig 1D-N). This phase marked the late maturation, where indicators such as vigor (Fig 1F-G, I-J) and longevity (Fig 1K-N) were optimized until the seed reached full maturity at stage R9.

Curiously, the anthocyanin results were entirely opposite ($PC_1$: 87.3%) to the acquisition of seed quality attributes during development (Fig 1 and Fig 2A). The results revealed a remarkably ordered pattern indicating differentiated levels of anthocyanin in dry seeds according to the maturation stages (Fig 2A). The anthocyanin index was high in the early stages (R5 and R6) and decreased significantly as the seed advanced to the late ones (R7, R8 and R9) (Fig 2B). This behavior coincided with the observed natural dehydration (70−30%) and resulted in a positive correlation (0.87) between anthocyanin and water content (Fig 2C). All other variables such as dry weight, desiccation tolerance, t50, seedling length, shoot and root dry weight of seedlings, germination of aged seeds, seedling emergence speed, seedling emergence, plant establishment in the field, shoot dry weight of plant stand, germination of stored seeds and normal seedlings of stored seeds showed a negative correlation with anthocyanin levels (ranging from −0.92 to −0.77). Therefore, anthocyanin abundance is strongly associated with the early stages of seed development (filling phase) and negatively correlates with traits associated with seed longevity (Figs 1 and 2).

### RNA-seq and flavonoid biosynthesis

To identify molecular mechanisms underpinning the anthocyanin dynamics during maturation phases, we performed RNA-seq analysis (see methods) of all development stages of peanut seeds. Principal component analysis (Fig 3A) showed

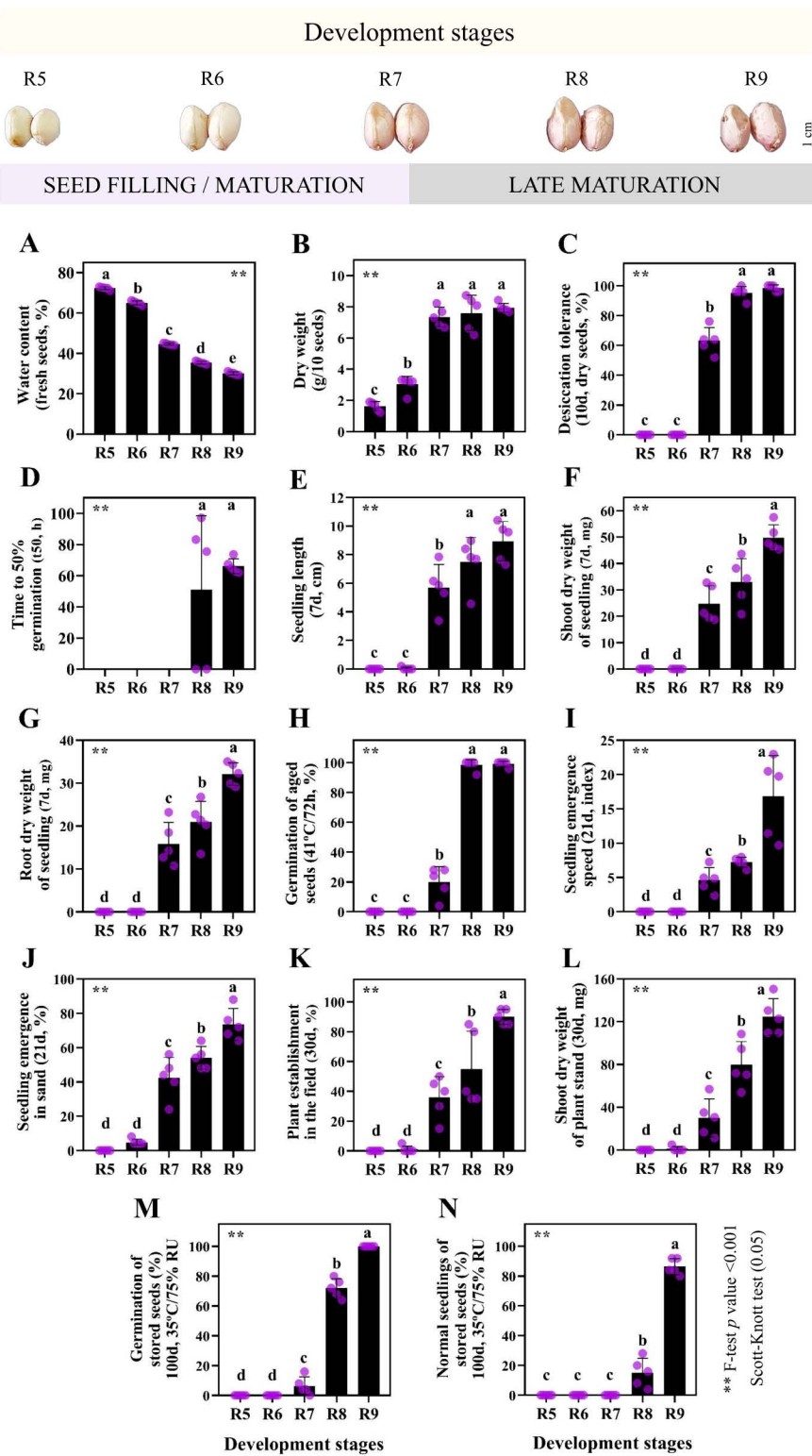

**Fig 1. Seed physiology during peanut development stages (cultivar IAC OL3).** (A-N) All means have standard deviations. Different letters on the bars are significantly different from each other by Scott Knott test (p-value ≤ 0.05). Asterisks (**) in all graphs signify the result of 1% significance by F-test (ANOVA). The dots along the bars are the biological replicates for each seed test.

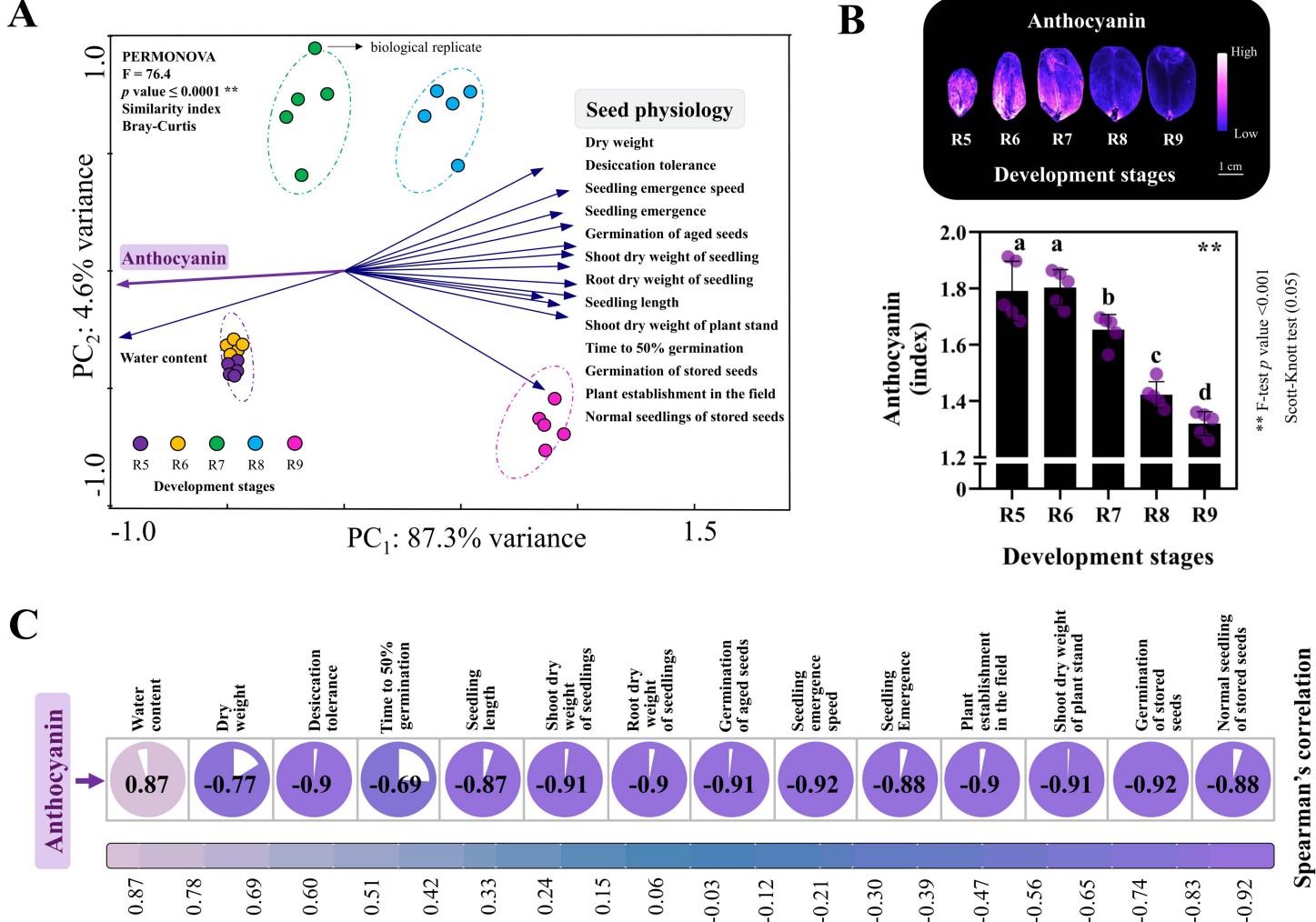

**Fig 2. Analysis performed with all seed physiology and anthocyanin data.** (A) Principal component analysis (PCA). (B) Anthocyanin index obtained by multispectral imaging. (C) Spearman correlation between anthocyanin and seed quality components.

separation between the early (R5 and R6) and late seed stages (R7, R8 and R9). The variability associated with principal component 1 of the transcriptome data (Fig 3A; $PC_1$: 91%) was similar to that obtained from seed physiology (Fig 2A; $PC_1$: 87.3%), demonstrating both these domains vary significantly during peanut seed development. Based on this result, differentially expressed genes were calculated between the early and late seed stages providing 8,854 significantly differentially expressed genes, of which 7,028 were upregulated and 1,826 were downregulated between the early and late stages, respectively (Fig 3B). The list of significant genes associated with the RNA-seq component of this work is fully available in the supplementary material (S1 File in S1 File).

We then queried our DEGs for overrepresented biological activities using gene ontology. Significantly enriched biological processes included cell wall formation, lipid biosynthesis, chlorophyll biosynthesis and hormone regulation (Fig 3C). In particular, one of the most enriched processes was flavonoid biosynthesis, (GO: 0009813) highlighting the moderation of anthocyanin biosynthesis (Fig 3C). Higher gene expression was observed for a chalcone isomerase (*AhCHI*), chalcone synthase (*AhCHS*), flavonol synthase (*AhFLS*), anthocyanidin reductase (*AhANR*) and leucoanthocyanidin reductase

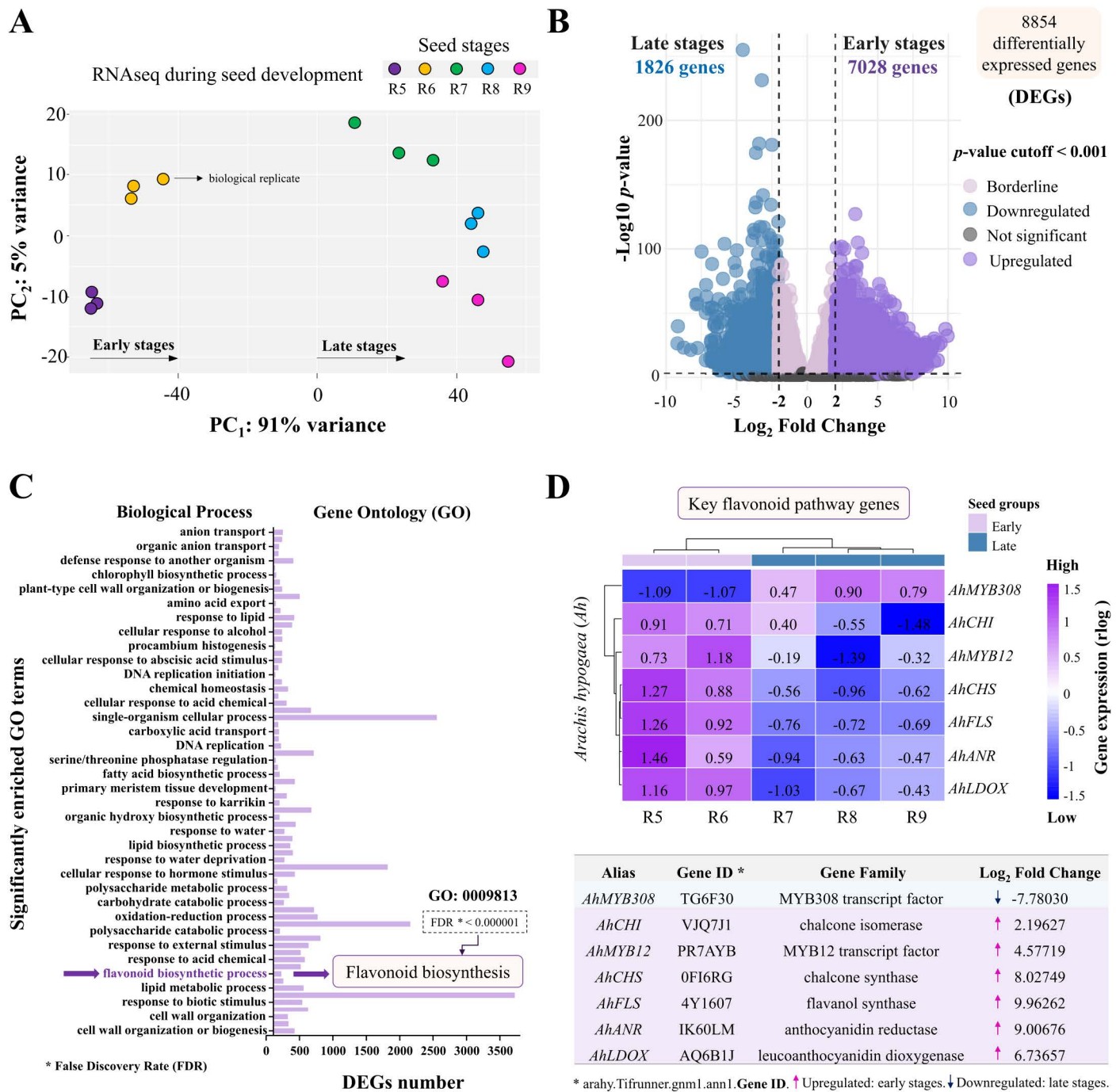

**Fig 3. RNA-seq analysis focused on flavonoid biosynthesis.** (A-B) PCA of transcriptome data and plotted to track significantly expressed genes from early and late stages. (C) Gene ontology (GO) analysis shows the groups of differently expressed genes (DEGs) in each enriched GO analysis, biological process. (D) Heatmap of genes associated with anthocyanin synthesis and the list of key genes chosen for RT-qPCR study. Gene expression data in the heat map were transformed using the DESeq2 regularized log (rlog) method, which stabilizes variance across mean expression levels.

(*AhLOX*) at stages R5 and R6. The transcription factor (TF) *AhMYB12* exhibited increased expression in the early stages and reduced expression in the late stages. Conversely, TF *AhMYB308* was only highly expressed in the late maturation stages (Fig 3D).

## RT-qPCR: Validation of genes during seed development

We identified increased expression in immature seeds (R5 and R6) of key genes involved in flavonoid biosynthesis throughout the different parts of the pathway: the beginning (*AhCHS* and *AhCHI*), middle (*AhFLS*) and end (*AhLOX* and *AhANR*). We validated the expression of these genes using qPCR and confirmed that their relative expression ($\Delta\Delta$Ct) was significantly higher at the early stages of seed development (Fig 4A-E). The transcription factors *AhMYB12* and *AhMYB308* were identified as potential regulators, as an activator ("on") and repressor ("off") respectively, of these flavonoid genes (Fig 4). Expression of *AhMYB12* was also higher at R5 and R6 (Fig 4F) whilst the expression values of *AhMYB308* were quite low at these stages and higher at later ones with a relative expression peak at R7 (Fig 4G). Finally,

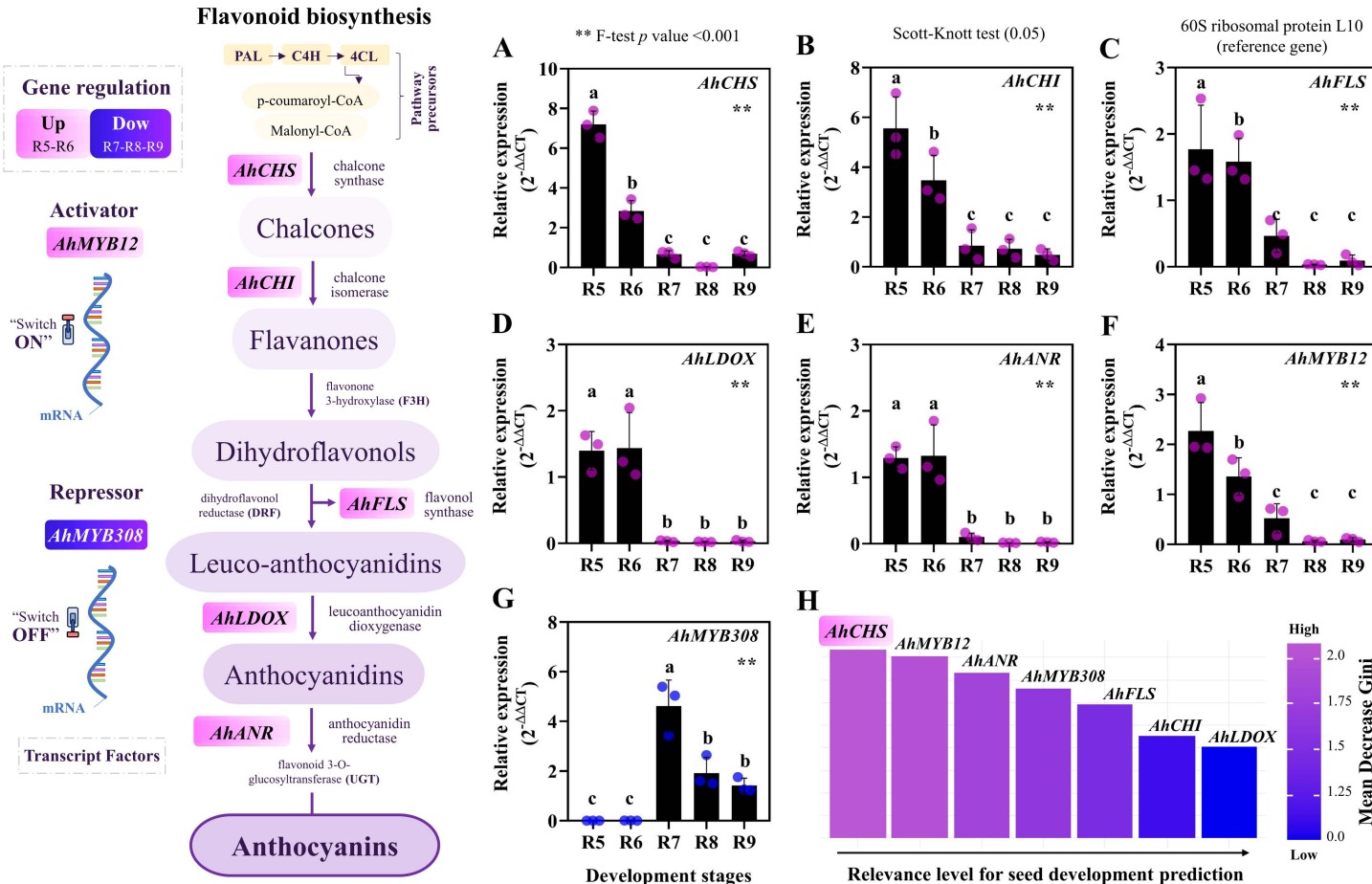

**Fig 4. Flavonoid biosynthesis pathway and the key genes along it.** (A-G) Relative gene expression by RT-qPCR during seed development stages. All means have standard deviations. Different letters on the bars are significantly different from each other by Scott Knott test (p-value ≤ 0.05). Asterisks (**) in all graphs signify the result of 1% significance by F-test (ANOVA). The dots along the bars are the biological replicates for each seed test. (H) Average Gini decrease performed to highlight the most relevant gene for predicting peanut seed development, *AhCHS*.

*AhCHS* was classified, based on Mean Decrease Gini, as the most relevant gene for predicting peanut seed development (Fig 4H).

## Genes associated with secondary metabolism in late maturation

In the GO analysis, within the flavonoid biosynthesis significant processes, a total of 14 downregulated genes were found. This negative regulation means that these genes were more expressed in the late stages (R7, R8 and R9). They were: UDP-glycosyltransferase 83A1, stilbene synthase 3-like, crocetin glucosyltransferase, beta-glucosidase 10, beta-glucosidase 11, hydroxyisourate hydrolase, pathogen-inducible salicylic acid glucosyltransferase, putative UDP-rhamnose:rhamnosyltransferase 1, linamarin synthase 2, putative stilbene synthase 2, Senescence-Related Gene 1, cryptochrome-1, vestitone reductase and UDP-glycosyltransferase 71K2. Interestingly, all of these had some documented relationship to the secondary metabolism (Table 1), which indicated a defensive protective strategy as the seed progressed towards late maturation.

## Key players in seed maturation phases

Due to our results suggesting that expression levels of *AhCHS* was the most important predictor of seed development (Fig 4H), the link between expression of *AhCHS* and other key players, as determined by literature review, involved in seed maturation (Table 2) was established in our RNA-seq data by correlation analysis (Fig 5). At early stages, correlations ranging from 0.74 to 0.95 were found between *AhCHS* and the following groups: 1) abscisic acid response (*ABI5* and *ABI3*); 2) B3 family (*REM23*, *FUS3* and *DOG1*); 3) gibberellin and auxin (*GA20ox1* and *AGR2*); 4) LEA proteins (*LEA3* and *LEA2*); 5) raffinose family oligosaccharide (*GSGT5*); and 6) heat shock proteins (*HSP18*, *HSFA2*, *HSP70* and *HSP21*). The similar transcriptional dynamics of *AhCHS* to these genes indicates that their upregulation is also prioritized in the seed filling phase (early stages). In contrast, at later stages, *AhCHS* expression correlations ranging from −0.63 to −0.84 were identified in the following gene groups: 7) LEA (*LEA5*, *LEA14*, *LEA29* and *LEA34*); 8) heat shock (*HSP19*,

**Table 1. The toolkit for resilience in peanut seed during late maturation. Downregulated genes associated with secondary metabolism found in the RNA-seq (Flavonoid biosynthesis, GO: 0009813).**

| Gene ID [1] | Descriptions [2] | Alias | Log$_2$ Fold Change [3] | Biological function | Literature |
|---|---|---|---|---|---|
| R5P48D | UDP-glycosyltransferase 83A1 | *UGT83A1* | −8.7478 | Defense responses | [98] |
| GBEE9Q | UDP-glucose iridoid glucosyltransferase | *UGT85A24* | −6.4866 | Resilience responses, iridoids | [99] |
| 12FNT9 | Crocetin glucosyltransferase | *UGT75L6* | −4.5942 | Antioxidant | [100] |
| G9Z134 | beta-glucosidase 10 | *BGLU10* | −4.5652 | Chemical defense, R2R3-MYB | [101] |
| 2A3E3X | beta-glucosidase 11 | *BGLU11* | −3.6176 | Chemical defense, phytoanticipins | [102] |
| QQMN09 | hydroxyisourate hydrolase | *HIUHase* | −3.1856 | Ureide pathway, nitrogen | [103] |
| 65UHA6 | pathogen-inducible salicylic acid glucosyltransferase | *SGT1* | −3.1065 | Pathogen control | [104] |
| Q8XZQ6 | putative UDP-rhamnose:rhamnosyltransferase 1 | *RRT1* | −3.0748 | Structural cell wall, pectin | [105] |
| Y6LS2L | linamarin synthase 2 | *UGT85K5* | −2.3463 | Chemical defense, cyanogenic | [106,107] |
| YS6PIM | putative stilbene synthase 2 | *STS2* | −2.3463 | Antioxidant, resveratrol | [78,90,108] |
| ZH3HJS | Senescence-Related Gene 1 | *SRG1* | −2.2242 | Self-immunity | [109,110] |
| VS4P4P | cryptochrome-1 | *CRY1* | −2.1990 | Thermotolerance, photoreceptor | [111,112] |
| K47810 | vestitone reductase | *VR* | −2.1856 | Antimicrobial | [113] |
| GYU2ZH | UDP-glycosyltransferase 71K2 | *UGT71K2* | −2.1312 | Stress tolerance | [114] |

[1]arahy.Tifrunner.gnm1.ann1.Gene ID (Phytozome).

[2]Phytozome (https://phytozome-next.jgi.doe.gov/) and NCBI (https://blast.ncbi.nlm.nih.gov/Blast.cgi) for *A. hypogaea*

[3]Downregulated genes in the late stages (R7, R8 and R9) according to DEG construction (Early x Late).

**Table 2. Key molecular players during peanut seed maturation phases from RNA-seq data.**

| Gene ID [1] | Descriptions[2] | Alias | Log$_2$ Fold Change[3] | Biological functions | Literature |
|---|---|---|---|---|---|
| **Early Stages – Seed water content is about 70–65%** | | | | | |
| 63C1KK | abscisic acid-insensitive 5-like protein 1 | *ABI5* | 6.8163 | Master regulator of LEA, RFO, HSP | [40] |
| 4K8YXT | B3 domain-containing transcription factor ABI3 | *ABI3* | 2.0705 | LEA regulation, seed maturation | [87] |
| 108MS0 | B3 domain-containing protein REM23 | *REM23* | 6.3784 | early seed development aspects | [85,86] |
| D9C186 | B3 domain-containing transcription factor FUS3 | *FUS3* | 3.8649 | seed development, lipid metabolism | [115,116] |
| 0HHX3D | Delay of Germination protein-like 4 | *DOG1* | 2.1621 | dormancy induction, seed maturation | [17] |
| 4TCZ1X | gibberellin 20 oxidase 1 | *GA20Ox1* | 6.9478 | gibberellin biosynthesis, seed development | [117,118] |
| 03QWZ4 | indole-3-acetic acid-induced protein | *ARG2* | 2.2970 | auxin homeostasis, seed development | [119,120] |
| EHR7LK | late embryogenesis abundant protein | *LEA3* | 6.6032 | protein stabilization, molecular protection | [16,72] |
| L9LT8B | late embryogenesis abundant protein | *LEA2* | 2.1096 | | |
| DBZB80 | galactinol--sucrose galactosyltransferase 5 | GSGT5 | 6.84.51 | raffinose, protect cell membranes | [18,19] |
| BM3N3L | 18.1 kDa class I heat shock protein | *HSP18* | 4.4017 | chaperone activity, protect membranes from desiccation, thermotolerance | [38,39,89] |
| ECZ3V7 | heat shock 70 kDa protein | *HSP70* | 2.6403 | | |
| ABBA5Z | heat shock transcription factor A-2 | *HSFA2* | 2.5362 | | |
| QR3Y2J | small heat shock protein, chloroplastic | *HSP21* | 2.0982 | | |
| **Late Stages – Seed water content is about 45–30%** | | | | | |
| 1PU3AF | late embryogenesis abundant protein | *LEA5* | −3.1330 | chaperone-like functions, proteins and membranes stabilization | [16,121] |
| 5K3X9F | desiccation protectant protein Lea14 | *LEA14* | −3.2242 | | |
| F8TXPA | Lea protein D-29 | *LEA29* | −2.0009 | | |
| LY00QK | Lea protein D-34 | *LEA34* | −5.0829 | | |
| V2AG0S | 18.5 kDa class I heat shock protein | *HSP19* | −2.3871 | chaperone-like functions, seed longevity | [10,11] |
| Y4YZQQ | 26.5 kDa heat shock protein, mitochondrial | *HSP27* | −2.0878 | | |
| 2BYT27 | heat shock 70 kDa protein, mitochondrial | *HSP70* | −2.433 | | |
| 1YYJ7Z | galactinol--sucrose galactosyltransferase | *GSGT* | −4.5010 | raffinose biosynthesis | [122,123] |
| B363MR | sucrose-phosphate synthase 2 | *SPS2* | −3.0305 | sucrose biosynthesis, sugar transporters | [91,92] |
| 6J07HI | bidirectional sugar transporter | *SWEET6b* | −2.2265 | | |
| S5JADZ. | sugar transport protein 10 | *STP10* | −4.2315 | | |
| XH4U3Y | Phenylalanine ammonia-lyase | *PAL* | −2.4316 | Secondary metabolism | [124,125] |
| YZ4W1Q | stilbene synthase 3-like | *STS3* | −6.2299 | Antioxidant, ROS scavengers | [78,90] |
| AS50A0 | glutathione reductase, cytosolic | *GSH* | −2.0845 | Antioxidant, ROS scavengers, longevity | [126,127] |

[1] arahy.Tifrunner.gnm1.ann1.Gene ID (Phytozome).

[2] Upregulated: Early stages (R5 and R6); Downregulated: late stages (R7, R8 and R9).

[3] Phytozome (https://phytozome-next.jgi.doe.gov/) and NCBI (https://blast.ncbi.nlm.nih.gov/Blast.cgi) for *A. hypogaea*

*HSP27* and *HSP70*); 9) sugar metabolism and transport (*GSGT*, *SPS2*, *SWEET6b* and *STP10*); 10) phenylpropanoid biosynthesis (*PAL*); and 11) antioxidant enzymes (*STS3* and *GR*). Clearly, other relevant genes showed prominent regulation during the late maturation phase, when *AhCHS* expression was minimized and seeds acquired the ability to survive in the dry state (Fig 5A-B).

## Discussion

### Linking anthocyanin dynamics with peanut seed development

Anthocyanin levels differ according to the seed water dynamics (Fig 2C). It was observed that high water content is favorable for anthocyanin storage in cell vacuoles and, upon dehydration, this accumulation is maintained in the dry weight of

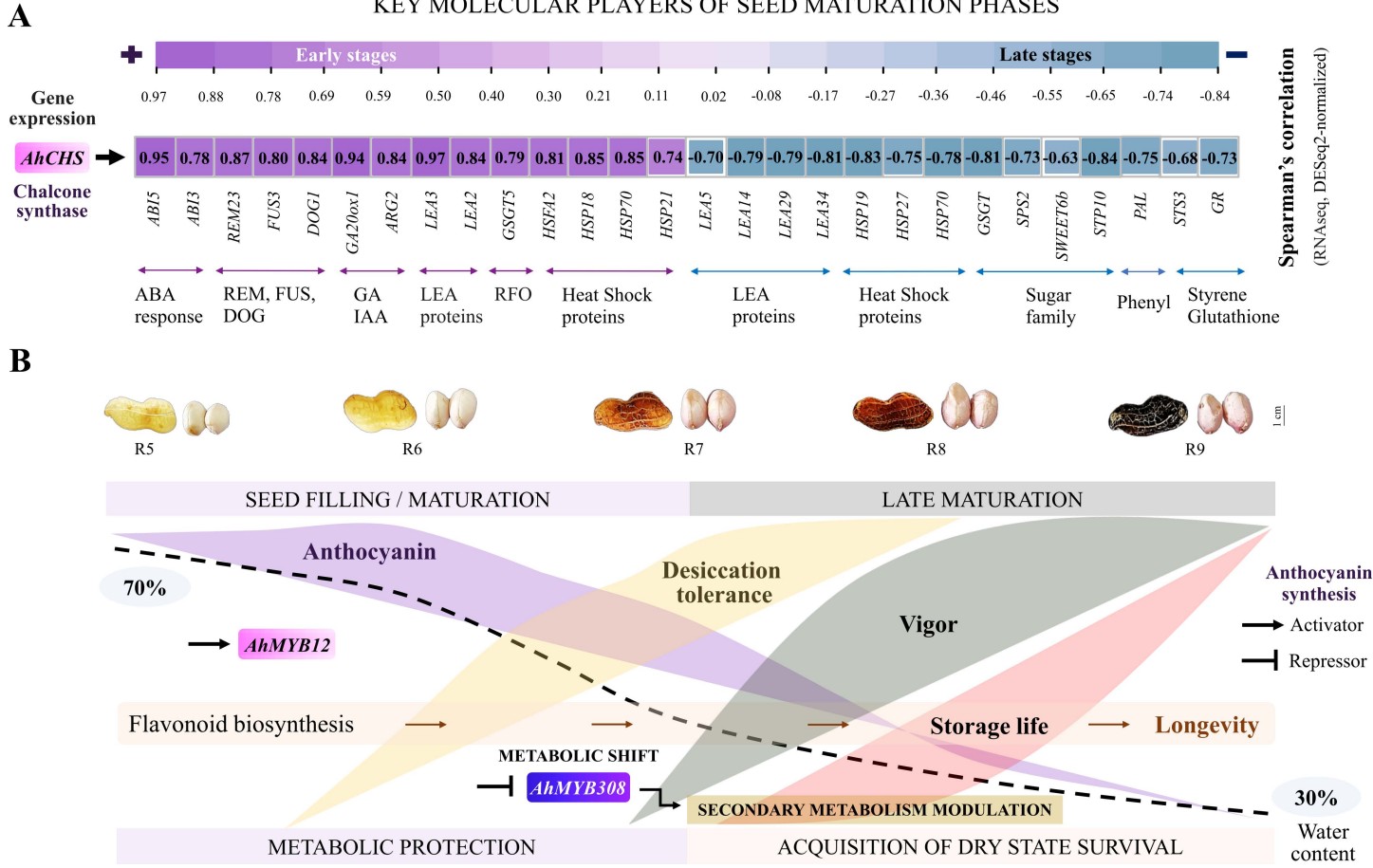

**Fig 5. Peanut seed development overview: physiological and molecular approach.** (A-B) Gene expression was obtained by gene counts normalized (DESeq2 size factor method), which adjusts for differences in sequencing depth between samples. The peanut fruit and seed images are original (cultivar IAC OL3).

the tissue (Fig 2B) [62,63]. During orthodox seed maturation, humidity conditions of 70–65% are commonly found during the early stages (Fig 1A). These stages are characterized by intense cellular changes, such as cell division, which occur under high water demand [64–66]. Additionally, seed filling is also intense in the early stages, where the high metabolic activity (i.e., respiration) is capable of generating excess free radicals that cause damage to cells [15,67]. Anthocyanins could adjust toxic oxidative levels, contributing to the deposition of reserves within a metabolically balanced framework of the seed [5,68–70]. This is especially important given that peanut seeds at early stages have high levels of $H_2O_2$ and MDA [43]. Conversely, at the late stages, there are agents such as *LEAs* and *HSPs* gene group, whose functions extend beyond those flavonoids (Fig 5A and Table 2) [11,16]. One example is the stabilization of proteins and cellular membranes at the molecular level under lower seed water content (45–30%) [16,71,72]. Probably because of this, the anthocyanin enrichment that is maintained at the beginning of maturation is less required at the end of seed development (Fig 2B). It is reasonable to suggest that flavonoid regulation appears to prepare the most immature seeds to acquire an appropriate chemical composition (Fig 1B), while gradually tolerating desiccation (Fig 1C). The crucial reserves such as proteins (LEAs and HSPs), sugars (ROFs), oil and minerals (K and Ca) are essential for cellular self-defense under seed water

changes towards the late stages [43,69]. This could mitigate potential cellular stressors, acting as a "metabolic guard" so that the maturation phase (seed filling) and the shift to late maturation may occur under homeostasis [24].

## Flavonoid biosynthesis: Transcriptional changes throughout seed development

The enriched flavonoid biosynthesis in our GO analysis (Fig 3C) highlights the existence of transcriptional programs that may modulate defense mechanisms throughout peanut maturation [73–75]. Interestingly, seed anthocyanin accumulation was aligned with the upregulation of key genes in its biosynthesis pathway, suggesting that both events were biologically correlated (Fig 2B, 3D). AhCHS was the key gene identified during seed development (Fig 4H), centrally positioned as a "bridge" in the pathway that produces other flavonoids involved in plant resistance [76,77]. It is important to note that this pathway is also mediated by induction/repression "switches" within the pathway (Fig 5) [31,33,36]; for instance, the transcription factor AhMYB12 is described as performing an essential role in flavonoid biosynthesis by positively regulating associated gene expression (Fig 3D) [32,33]. Indeed, its increased activity coinciding with increased gene expression suggests its transcriptional demand was required in early seed stages (Fig 3D) [30,31]. Curiously, in the late stages, most flavonoid genes showed decreased expression levels, while AhMYB308 exhibited increased transcription (Fig 4G). It's possible that it represses key flavonoid biosynthetic genes (Fig 3D) while increasing the availability of other precursors of secondary metabolism [32,34]. This flexible action of AhMYB308 may contribute to the establishment of a "stress protection toolkit" in the late stages (Table 1). The potential mechanisms in peanuts appear to function as a dynamic network of nuclear decisions, regulating transcriptional changes based on the demands of each maturation phase (Fig 3 and 4).

## The "Toolkit" for seed resilience: Exploring secondary metabolism during late maturation

We identified a set of genes upregulated exclusively during the late stages of peanut seed development (Table 1) which we hypothesize may equip the developing seed with a "toolkit" to ensure it is prepared for germinating and subsequent establishment [4,78]. The genes found include those associated with antioxidants, antifungals, cyanogen common in other species (e.g., Manihot esculenta Crantz), photoreceptors associated with seed dormancy, thermoprotectants, N metabolism contributors, cell wall structural components (e.g., lignin, pectin and suberin), insecticidal compounds and stress tolerance agents, which have all been documented (Table 1) [26,79]. An interesting aspect to this is that the implementation of this strategy takes place in an underground environment where the diversity of biotic and abiotic factors that can interfere in the seed development is vast [80,81]. Perhaps this singular habit of maturation under these potential stressors justifies the great embryonic protection established transcriptionally (Fig 3C). Furthermore, an important aspect of this protective enhancement is the seed's chemical composition, which is predominantly made up of lipids (48–50%), prone to oxidation [15,43,82]. Unsaturated fatty acids, such as linoleic acid, are significant sources of peroxidation, which demand robust enzymatic performance from the cellular machinery to protect it from lipid instability [83,84]. Thus, the peanut development appears to have evolved with a reinforcement of genetic configurations that provide robust support for the viability of its offspring. This enhanced late secondary metabolism may function as a sophisticated "toolkit", providing necessary resilience to: i) both biotic and abiotic stresses; ii) acquisition of dry-state survival; and iii) maximum protection during peanut seed storage life (Fig 1M-N).

## Key players in peanut seed development: The birth of longevity in the dry state

We identified distinct groups of genes that are preferentially expressed during the different stages of the peanut seed's journey towards full maturity (Table 2). Flavonoid biosynthesis appears to be one of the gears in a sophisticated set of mechanisms that contribute to the molecular events required for proper seed development (Fig 5) [79,85,86]. Its workflow in the beginning of peanut maturation coincides with the activity of master orchestrators of legume seed development (Fig 5A), such as ABA responses described in model plants [12,40]. These key players are associated with acquisition of desiccation tolerance, longevity and modulation of flavonoid biosynthesis through the MYB family of transcription factors [40,87,88].

Notably, flavonoids positively correlate with a diverse set of genes and transcription factors (e.g., *FUS3* and *HSFA2*) extending to regulators of lipid and carbohydrate synthesis, agents of hormonal balance, seed dormancy and chaperone protein synthesis (Fig 5, Table 2) [16,38,39,72,89]. It is possible that the activities of these genes are coordinated in synergy with the upregulation of anthocyanin pathway genes (Fig 4), as part of a shared natural effort to acquire seed quiescence [18,19].

In late maturation other agents such as chaperone proteins (e.g., HSP70), sugar transporters, phenylpropanoids and antioxidants were genetically orchestrated (Table 2) [78,90–92]. These protective mechanisms have been described as promoters of molecular stability throughout seed storage time, preserving vigor and ensuring the continuity of the species across different environmental scenarios [10,11,13,93]. What is interesting to consider is how peanut seeds are genetically prepared to survive in nature only if they experience late maturation, a phase in which maximum physiological quality is acquired (Fig 1) [11,24,94]. Flavonoid biosynthesis results from a network of transcriptional decisions in the cell nucleus, which could thus be considered to be the birthplace of seed storage life (Fig 5B). Thus, the correlation of flavonoid synthesis with the long survival of seeds in the dry state can be considered part of the path towards full peanut maturity (stage R9).

### Applications in agriculture and future approaches

Our work highlights some findings that may be of interest in an agricultural setting. The first is the use of technologies, based on multispectral images to detect seeds from the late stages by analyzing anthocyanin levels non-destructively, as a strategy for mitigating the uneven maturation inherent in peanut plants [24]. This could optimize the selection of seeds with superior longevity and enhanced bioprotection [8,9], since an increase in anthocyanin reflects embryonic immaturity during maturation phases (Fig 1-2). Furthermore, it enhances the efficiency of seed analysis compared to traditional methods, which are often subjective and require extensive training [8,9]. In the cellular context, assessing the expression levels of *AhCHS* would allow the evaluation of seed maturity as a proxy of the extent of flavonoid biosynthesis (Fig 3) [24,76,77]. This could become a potential tool for seed samples analysis, enabling real-time results through qPCR, adapted to industrial demands (Fig 4). Additionally, our findings are valuable for initiating classical plant breeding programs aimed at selecting peanut lines through molecular markers linked to the secondary metabolism (Table 1). This approach could lead to new varieties with enhanced resilience, improving resistance against different kinds of stresses [4–7]. From the food industry's perspective, new peanut-based products enriched with antioxidants could be developed, highlighting the nutritional benefits for human health [95]. Regarding future approaches, gene-editing projects using CRISPR-Cas9 could help refine our understanding of the mechanisms associated with the genes we identified in this study and reported in literature (Fig 4) [76,77,96]. These results provided may be useful for the biotechnological development of new peanut crops (Table 2), especially in future attempts to address challenges posed by extreme climate changes [97]. Thus, these findings can contribute to technological steps towards the constant improvement of the peanut chain in producing nations.

### Conclusions

The peanut has a unique physiology and biology among cultivated legume species. The involvement of flavonoid biosynthesis in seed development is suggested here as a contributor to its resilience during the acquisition of physiological quality attributes. The associated transcriptional changes implement secondary metabolism responses adapted to each maturation phase towards full seed maturity. We present a potential mechanism in peanut that contributes to seed survival in the dry state, with maximum storage life and enhanced bioprotection.

### Supporting information

**S1 File. S1 Table.** Quality control of RNA samples extracted from fresh peanut seeds. **S2 Table.** Reads mapped to the peanut genome. **S3 Table.** Primers designed for RT-qPCR study. **S1 File.** Significant genes associated with the RNA-seq. (ZIP)

## Acknowledgments

We are grateful to Roger Hutchings for reviewing the English of the manuscript. We also thank the University of Warwick (School of Life Sciences, UK) and Professor George Bassel for their valuable support in this work.

## Author contributions

**Conceptualization:** Gustavo Fonseca de Oliveira, Edvaldo Aparecido Amaral da Silva.

**Data curation:** Gustavo Fonseca de Oliveira, Edvaldo Aparecido Amaral da Silva.

**Formal analysis:** Gustavo Fonseca de Oliveira, Liam Walker, Rômulo Pedro Macêdo Lima.

**Funding acquisition:** Edvaldo Aparecido Amaral da Silva.

**Investigation:** Gustavo Fonseca de Oliveira.

**Methodology:** Gustavo Fonseca de Oliveira, Liam Walker, Rômulo Pedro Macêdo Lima, Thiago Barbosa Batista, Clíssia Barboza Mastrangelo, Edvaldo Aparecido Amaral da Silva.

**Project administration:** Edvaldo Aparecido Amaral da Silva.

**Supervision:** Edvaldo Aparecido Amaral da Silva.

**Validation:** Gustavo Fonseca de Oliveira, Liam Walker, Thiago Barbosa Batista, Clíssia Barboza Mastrangelo.

**Visualization:** Gustavo Fonseca de Oliveira, Liam Walker, Rômulo Pedro Macêdo Lima, Thiago Barbosa Batista, Clíssia Barboza Mastrangelo, Edvaldo Aparecido Amaral da Silva.

**Writing – original draft:** Gustavo Fonseca de Oliveira.

**Writing – review & editing:** Liam Walker, Thiago Barbosa Batista, Clíssia Barboza Mastrangelo, Edvaldo Aparecido Amaral da Silva.

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
