## [Decision Letter · Decision Letter 0]

PONE-D-25-12406Towards Resilience: Transcriptional Insights on Flavonoid Biosynthesis During Peanut Seed Maturation PhasesPLOS ONE

Dear Dr. Fonseca de Oliveira,

Thank you for submitting your manuscript to PLOS ONE. After careful consideration, we feel that it has merit but does not fully meet PLOS ONE’s publication criteria as it currently stands. Therefore, we invite you to submit a revised version of the manuscript that addresses the points raised during the review process.

Based on the reviewers' feedback, the manuscript can be accepted after all the reviewers' comments have been addressed.

I kindly request you to submit a revised version of your manuscript after addressing all the comments. Please note that the comments from Reviewer 2 are annotated within the manuscript PDF.

I look forward to receiving your revised manuscript.

We look forward to receiving your revised manuscript.

Kind regards,

Koppolu Raja Rajesh Kumar, PhD

Academic Editor

PLOS ONE

Journal Requirements:

2. Please amend your list of authors on the manuscript to ensure that each author is linked to an affiliation. Authors’ affiliations should reflect the institution where the work was done (if authors moved subsequently, you can also list the new affiliation stating “current affiliation:….” as necessary).

3. Your abstract cannot contain citations. Please only include citations in the body text of the manuscript, and ensure that they remain in ascending numerical order on first mention.

Reviewers' comments:

Reviewer's Responses to Questions

**Comments to the Author**

1. Is the manuscript technically sound, and do the data support the conclusions?

Reviewer #1: Yes

Reviewer #2: Yes

2. Has the statistical analysis been performed appropriately and rigorously? 

Reviewer #1: Yes

Reviewer #2: Yes

3. Have the authors made all data underlying the findings in their manuscript fully available?

Reviewer #1: Yes

Reviewer #2: Yes

4. Is the manuscript presented in an intelligible fashion and written in standard English?

Reviewer #1: Yes

Reviewer #2: Yes

5. Review Comments to the Author

Reviewer #1: The manuscript "Towards Resilience: Transcriptional Insights on Flavonoid Biosynthesis During Peanut Seed Maturation Phases" presents a comprehensive investigation into the role of flavonoids, particularly anthocyanins, in peanut seed maturation. The study integrates physiological assessments with transcriptomic analyses to explore the dynamic regulation of flavonoid biosynthesis and its association with seed resilience. Below are my comments and suggestions for improvement:

1. Mechanistic Insights: While the study links flavonoid biosynthesis to seed resilience, the exact mechanism by which flavonoids interact with other protective pathways (e.g., LEA proteins, HSPs) remains unclear. Direct measurements of reactive oxygen species (ROS) or oxidative damage could strengthen the proposed antioxidant role of anthocyanins.

2. Gene Annotation and Formatting:

Some gene IDs in Tables 1 and 2 lack clarity (e.g., "AhCHS (0F16RG)"). Ensure consistent use of standardized gene nomenclature (e.g., AhCHS) and italicization for gene symbols.

The description of transcription factors (e.g., AhMYB12 as an activator and AhMYB308 as a repressor) needs experimental validation (e.g., ChIP-seq or promoter-binding assays) to support regulatory roles.

3. Figure Accessibility: The figures are referenced but not visible in the provided text. Ensure high-resolution images with clear labels and legends are included, especially for PCA plots and heatmaps.

4. Terminology Consistency: Use "RNA-seq" instead of "RNAseq" throughout the text. Ensure consistent formatting of gene names (e.g., ABI3 vs. ABI5).

5. Discussion Expansion:

Elaborate on the feasibility of multispectral imaging for industrial seed sorting. How does this compare to existing methods?

The mention of CRISPR applications, while intriguing, feels tangential. Consider focusing on immediate implications or briefly contextualize it as a future direction.

6. Abstract: Replace "RNAseq" with "RNA-seq."

7. Methods: Clarify the number of biological replicates for RNA-seq (stated as three replicates per stage) and ensure this aligns with the DESeq2 analysis.

8. References: Verify citations for accuracy (e.g., Ref 28 cites "Peanuts: Genetics, Processing, and Utilization" but lists the publication year as 2016, while Ref 27 refers to USDA data from "Dec 2025," which appears to be a typo).

Reviewer #2: 1. The author explored the possible links between flavonoids and the coordinators of peanut seed development, and showed that flavonoids were transcriptionally involved in peanut seed development, acting systematically on secondary metabolism throughout the maturation phases.

2. The abstract of this paper, a little bit obscure, should be presented more accurately, concisely and clearly.

3. Please see the minor revisions in the attached PDF file directly.

6. PLOS authors have the option to publish the peer review history of their article (what does this mean? ). If published, this will include your full peer review and any attached files.

**Do you want your identity to be public for this peer review?** For information about this choice, including consent withdrawal, please see our Privacy Policy .

Reviewer #1: No

Reviewer #2: **Yes: ** Xiaorong Wan

---

## [Author Response · Author response to Decision Letter 1]

5 May 2025

Dear reviewers,

Please find below the answers to all comments made on the manuscript (PONE-D-25-12406)

Kind regards

Dr. Gustavo Roberto Fonseca de Oliveira

Reviewer #1:

The manuscript "Towards Resilience: Transcriptional Insights on Flavonoid Biosynthesis During Peanut Seed Maturation Phases" presents a comprehensive investigation into the role of flavonoids, particularly anthocyanins, in peanut seed maturation. The study integrates physiological assessments with transcriptomic analyses to explore the dynamic regulation of flavonoid biosynthesis and its association with seed resilience. Below are my comments and suggestions for improvement:

1. Mechanistic Insights: While the study links flavonoid biosynthesis to seed resilience, the exact mechanism by which flavonoids interact with other protective pathways (e.g., LEA proteins, HSPs) remains unclear. Direct measurements of reactive oxygen species (ROS) or oxidative damage could strengthen the proposed antioxidant role of anthocyanins.

Answer: We appreciate this important observation regarding the mechanistic interactions between flavonoids and other protective pathways. Indeed, while our study highlights a potential association between flavonoid biosynthesis and seed resilience, we recognise that the precise mechanisms—particularly the interplay with LEA proteins and HSPs (see results in the Table 2)—remain to be fully elucidated. Although we did not directly measure reactive oxygen species (ROS) or oxidative damage in this study, our interpretation is supported by previous literature that reports the antioxidant function of anthocyanins in peanut seeds during maturation and other species (please see the papers below). Additionally, the expression patterns observed in our dataset are consistent with a coordinated response involving stress-related genes, including those associated with protective proteins. However, in light of your valuable comments, we have added further arguments (see lines 348). Thank you for encouraging us to reflect more critically on this aspect.

43. Fonseca de Oliveira GR, Hirai WY, Ferreira DS, Silva KPOM da, Silva GC, Moraes TB, et al. Spectroscopy Technologies to Screen Peanut Seeds with Superior Vigor Through “Chemical Fingerprinting.” Agronomy. 2024; 14: 1-17. https://doi:10.3390/agronomy14112529

68. Cerqueira JVA, de Andrade MT, Rafael DD, Zhu F, Martins SVC, Nunes-Nesi A, et al. Anthocyanins and reactive oxygen species: a team of rivals regulating plant development? Plant Mol Biol. 2023; 112: 213–223. http://doi:10.1007/s11103-023-01362-4

70. Liu Y, Tikunov Y, Schouten RE, Marcelis LFM, Visser RGF, Bovy A. Anthocyanin biosynthesis and degradation mechanisms in Solanaceous vegetables: A review. Front Chem. 2018; 6. http://doi:10.3389/fchem.2018.00052

2. Gene Annotation and Formatting: Some gene IDs in Tables 1 and 2 lack clarity (e.g., "AhCHS (0F16RG)"). Ensure consistent use of standardized gene nomenclature (e.g., AhCHS) and italicization for gene symbols. The description of transcription factors (e.g., AhMYB12 as an activator and AhMYB308 as a repressor) needs experimental validation (e.g., ChIP-seq or promoter-binding assays) to support regulatory roles.

Answer: Tables 1 and 2 present the gene IDs along with footnotes to clarify the origin of the nomenclature (lines 881 and 888). We chose to include both pieces of information to facilitate readers' consultation of the Phytozome database. In the specific case mentioned, 0F16RG is a particular example of chalcone synthase (CHS) found in our peanut dataset. As there are numerous predicted CHS genes in the peanut genome, we opted to include both the alias AhCHS (a commonly used nomenclature) and the ID 0F16RG. This approach allows readers to more easily locate and compare results with other studies that refer to this specific CHS in peanuts using the same nomenclatures. The same rationale applies to the other genes presented. As suggested, gene symbols have been revised throughout the text and figures.

Regarding the transcription factors, they have already been characterised as repressors or activators in other plant species, in studies associated with flavonoid biosynthesis (please refer to the citations below, also included in the text). While this has not been experimentally validated in our peanut study, there is strong evidence in the literature that supports the descriptions provided. Based on our findings, future studies will be able to explore these observations in more detail and test specific hypotheses to better understand the regulatory roles of AhMYB12 and AhMYB308, which were beyond the scope of the present work. Therefore, our reasoning was based on previous literature as well as the behaviour observed in our data (Figure 3D and Figures 4F and 4G). We added the word 'potential' to the text when mentioning transcription factors in the results section, in reference to their potential roles as repressors and activators of flavonoid biosynthesis in peanut (line 300). We hope this explanation clarifies the point raised and we remain open to further discussion. Thank you very much for your thoughtful comments.

32. Mehrtens F, Kranz H, Bednarek P, Weisshaar B. The Arabidopsis transcription factor MYB12 is a flavonol-specific regulator of phenylpropanoid biosynthesis. Plant Physiol. 2005; 138: 1083–1096. doi:10.1104/pp.104.058032

33. Wang N, Xu H, Jiang S, Zhang Z, Lu N, Qiu H, et al. MYB12 and MYB22 play essential roles in proanthocyanidin and flavonol synthesis in red-fleshed apple (Malus sieversii f. niedzwetzkyana). Plant J. 2017; 90: 276–292. https://doi:10.1111/tpj.13487

34. Dhakarey R, Yaritz U, Tian L, Amir R. A Myb transcription factor, PgMyb308-like, enhances the level of shikimate, aromatic amino acids, and lignins, but represses the synthesis of flavonoids and hydrolyzable tannins, in pomegranate (Punica granatum L.). Hortic Res. 2022; 9: 1–13. https://doi:10.1093/hr/uhac008

36. LaFountain AM, Yuan Y. Repressors of anthocyanin biosynthesis. New Phytol. 2021; 231: 933–949. https://doi:10.1111/nph.17397

3. Figure Accessibility: The figures are referenced but not visible in the provided text. Ensure high-resolution images with clear labels and legends are included, especially for PCA plots and heatmaps.

Answer: We agree that the resolution of the figures appears low in the PDF file. However, all figures were submitted in high resolution (600 dpi) and are available for download via the link provided at the top of each figure page. We anticipate that, in the final version of the publication, the figures will be displayed as originally submitted. Thank you for your thoughtful observation.

4. Terminology Consistency: Use "RNA-seq" instead of "RNAseq" throughout the text. Ensure consistent formatting of gene names (e.g., ABI3 vs. ABI5).

Answer: The changes have been made as suggested. Thank you for your valuable input.

5. Discussion Expansion: Elaborate on the feasibility of multispectral imaging for industrial seed sorting. How does this compare to existing methods? The mention of CRISPR applications, while intriguing, feels tangential. Consider focusing on immediate implications or briefly contextualize it as a future direction.

Answer: We agree with the comment and new arguments have been incorporated into the Discussion section as suggested. Thank you for your insightful recommendation (445-460).

6. Abstract: Replace "RNAseq" with "RNA-seq."

Answer: The requested changes have been implemented as suggested. We appreciate your thoughtful feedback.

7. Methods: Clarify the number of biological replicates for RNA-seq (stated as three replicates per stage) and ensure this aligns with the DESeq2 analysis.

Answer: As suggested, we added the information “(three replicates of 15 fresh seeds, n=45)” to the Methods section to clarify the number of replicates used (lines 200 and 224 – Revised Manuscript with Track Changes). Thank you for your helpful suggestion.

8. References: Verify citations for accuracy (e.g., Ref 28 cites "Peanuts: Genetics, Processing, and Utilization" but lists the publication year as 2016, while Ref 27 refers to USDA data from "Dec 2025," which appears to be a typo).

Answer: Great observation. The corrections were made. Thanks.

Reviewer #2:

All the corrections suggested through the text (PDF file) were made (lines 24, 27, 31, 298, 303, 362 – Revised Manuscript with Track Changes). Thanks.

---

## [Decision Letter · Decision Letter 1]

Towards Resilience: Transcriptional Insights on Flavonoid Biosynthesis During Peanut Seed Maturation Phases

PONE-D-25-12406R1

Dear Dr. Fonseca de Oliveira,

We’re pleased to inform you that your manuscript has been judged scientifically suitable for publication and will be formally accepted for publication once it meets all outstanding technical requirements.

Kind regards,

Koppolu Raja Rajesh Kumar, PhD

Academic Editor

PLOS ONE

Additional Editor Comments (optional):

Reviewers' comments:

Reviewer's Responses to Questions

**Comments to the Author**

1. If the authors have adequately addressed your comments raised in a previous round of review and you feel that this manuscript is now acceptable for publication, you may indicate that here to bypass the “Comments to the Author” section, enter your conflict of interest statement in the “Confidential to Editor” section, and submit your "Accept" recommendation.

Reviewer #1: All comments have been addressed

Reviewer #2: All comments have been addressed

2. Is the manuscript technically sound, and do the data support the conclusions?

Reviewer #1: Yes

Reviewer #2: Yes

3. Has the statistical analysis been performed appropriately and rigorously? 

Reviewer #1: Yes

Reviewer #2: Yes

4. Have the authors made all data underlying the findings in their manuscript fully available?

Reviewer #1: Yes

Reviewer #2: Yes

5. Is the manuscript presented in an intelligible fashion and written in standard English?

Reviewer #1: Yes

Reviewer #2: Yes

6. Review Comments to the Author

Reviewer #1: (No Response)

Reviewer #2: (No Response)

7. PLOS authors have the option to publish the peer review history of their article (what does this mean? ). If published, this will include your full peer review and any attached files.

**Do you want your identity to be public for this peer review?** For information about this choice, including consent withdrawal, please see our Privacy Policy .

Reviewer #1: No

Reviewer #2: **Yes: ** Wan Xiaorong

---

## [Editor Report · Acceptance letter]

PONE-D-25-12406R1

PLOS ONE

Dear Dr. Fonseca de Oliveira,

I'm pleased to inform you that your manuscript has been deemed suitable for publication in PLOS ONE. Congratulations! Your manuscript is now being handed over to our production team.

Kind regards,

on behalf of

Dr. Koppolu Raja Rajesh Kumar

Academic Editor

PLOS ONE